# Examining Influenza Vaccination Patterns and Barriers: Insights into Knowledge, Attitudes, and Practices among Diabetic Adults (A Cross-Sectional Survey)

**DOI:** 10.3390/vaccines11111689

**Published:** 2023-11-03

**Authors:** Walid Al-Qerem, Anan Jarab, Abdel Qader AlBawab, Alaa Hammad, Badi’ah Alazab, Daoud Abu Husein, Judith Eberhardt, Fawaz Alasmari

**Affiliations:** 1Department of Pharmacy, Faculty of Pharmacy, Al-Zaytoonah University of Jordan, Amman 11733, Jordan; abdelqader.albawab@zuj.edu.jo (A.Q.A.); alaa.hammad@zuj.edu.jo (A.H.); b.alazab@zuj.edu.jo (B.A.); d.abuhusein@zuj.edu.jo (D.A.H.); 2Department of Clinical Pharmacy, Faculty of Pharmacy, Jordan University of Science and Technology, Irbid 22110, Jordan; asjarab@just.edu.jo; 3College of Pharmacy, Al Ain University, Abu Dhabi 112612, United Arab Emirates; 4School of Social Sciences, Humanities and Law, Department of Psychology, Teesside University, Borough 15 Road, Middlesbrough TS1 3BX, UK; j.eberhardt@tees.ac.uk; 5Department of Pharmacology and Toxicology, College of Pharmacy, King Saud University, Riyadh 12372, Saudi Arabia; ffalasmari@ksu.edu.sa

**Keywords:** diabetes, complications, influenza, influenza vaccine, morbidity

## Abstract

Diabetes mellitus is a prevalent global chronic condition affecting individuals of all ages. People with diabetes face an elevated risk of lower respiratory tract infections such as pulmonary tuberculosis, influenza, and pneumonia. Additionally, the influenza virus increases the likelihood of deep venous thrombosis and pulmonary embolism. This study examined the knowledge, attitudes, and practices of diabetic patients regarding the influenza vaccine. This study involved 418 diabetic patients (53.3% female) at Jordanian outpatient respiratory clinics, with an average age of 49 (±14) years. The results showed that 70.6% had never received the influenza vaccine, and only 23.7% intended to do so in the current year. A positive attitude toward the influenza vaccine significantly reduced hesitancy to get vaccinated (OR = 0.505, 95% CI 0.424–0.601, *p* < 0.001). The duration of diabetes exhibited a positive association with vaccine hesitancy (OR = 1.053, 95% CI 1.006–1.102, *p* = 0.028). The primary reason for not getting vaccinated was a lack of awareness of its benefits (42.6%). Future health education programs should emphasize the importance of the influenza vaccine for diabetic patients and address their concerns.

## 1. Introduction

According to the World Health Organization (WHO), diabetes is a chronic disease of elevated glucose levels in the blood that may lead to serious complications and damage to different body organs if not properly controlled. [1] In the last thirty years, the prevalence of Type 2 diabetes (DM), which is associated with insulin resistance, has risen dramatically in countries of different income levels [1]. Type 1 DM (also known as juvenile diabetes or insulin-dependent diabetes) is a chronic condition in which the pancreas produces little or no insulin [1]. According to the National Center for Diabetes, Endocrinology, and Genetics, the prevalence of DM and pre-DM in Jordan is around 45% [2]. In addition, DM prevalence in Jordan is higher than in other countries when compared globally and in the Middle East [2]. Almost half of diabetic patients are unaware of their disease and are thus more prone to developing diabetic complications, which are more common in Type I diabetes [3].

Several studies have reported an increased risk of lower respiratory tract infections such as pulmonary tuberculosis, influenza, and pneumonia in patients with DM [4]. In addition, the influenza virus increases the risk of deep venous thrombosis and pulmonary embolism [5]. According to the latest data published by WHO in 2020, influenza and pneumonia caused 3.59% of total deaths in Jordan [6].

Influenza is a vaccine-preventable disease, and influenza vaccination is generally recommended for all persons with DM [5]. A study conducted in Jordan in 2019 concluded that Jordanian older adults had negative attitudes toward receiving an influenza vaccine [7]. Furthermore, a cross-sectional Jordanian survey in 2022 involving 564 individuals aged 18–64 with chronic diseases, 71% of whom had DM, showed a low uptake of the influenza vaccine among DM patients [8]. This hesitancy towards vaccination could be attributed to misconceptions of the influenza vaccine, similar to the misconceptions of the COVID-19 vaccine observed during the COVID-19 pandemic [9]. While there are potential adverse effects, such as myocarditis and pericarditis, associated with influenza vaccination [10], the advantages of getting vaccinated outweigh the risks, especially for at-risk groups, such as individuals with DM [10].

In Jordan, the influenza vaccine is available in community pharmacies to the public. It is not mandatory for any particular population group, including those with diabetes, and it is not provided free of charge. Additionally, it is not part of the national vaccination program. There are no official reports available on influenza vaccination coverage among Jordanians, and the country lacks awareness campaigns regarding the vaccine [11]. This highlights the need to comprehend the factors contributing to the low uptake of the influenza vaccine and the phenomenon of vaccine rejection/hesitancy. This understanding is essential for the planning, implementation, and evaluation of effective immunization programs in Jordan and may help healthcare providers develop health campaigns to increase vaccination against influenza among diabetics and thereby improve diabetes control.

Therefore, the current research aimed to explore diabetics’ knowledge about diabetes, influenza and its vaccine, their adherence to practices that improve diabetes control, and their attitudes and practices regarding the influenza vaccine, then evaluate the variables associated with their intention to receive the influenza vaccine in the current year.

## 2. Materials and Methods

### 2.1. Methodology

The present study was carried out with diabetic adult patients who attended outpatient clinics at two large Jordanian hospitals. The first is AlBashir Hospital, which is located in the capital city, Amman, and serves a substantial patient population from various regions. The second, King Abdullah University Hospital, which is located in the northern city of Irbid, serves patients from the northern governates of Jordan. These factors contributed to the representativeness of the study’s sample within the context of Jordan. Inclusion criteria involved adults 18 years or older who had been diagnosed with DM and were willing to participate in the study. Patients were interviewed by the research pharmacist and provided with a brief description of the aims of the study. Participants were informed about confidentiality, anonymity of the obtained data, and voluntary participation, and an informed consent form was signed by all participants. Out of 520 patients approached, 418 consented to participate in this study, resulting in a response rate of 80.4%. The data were collected between Jan and May 2023. This study adhered to the ethical guidelines outlined in the Declaration of Helsinki. Ethical approval was granted by Al-Zaytoonah University of Jordan (Reference #22/20/2022–2023), Jordan University of Science and Technology (Reference #2022/07), and the Ministry of Health in Jordan (Reference #MOH/REC/2023/119). We have followed the Consensus-Based Checklist for Reporting of Survey Studies (CROSS) [12]. 

### 2.2. Data Collection and Study Instruments

#### 2.2.1. Self-Modified Questionnaire

The data were collected with a questionnaire custom-designed using Google Forms. This questionnaire was developed after extensive literature review and was translated from English into Arabic following its development. The questionnaire consisted of six parts. The first part included demographic data of the patient, including age, gender, educational level, marital status, socioeconomic status, DM duration, DM type, HbA1c level, smoking, and passive smoking exposure. 

The second part assessed patients’ knowledge of DM (seven items), and eight items assessed knowledge of influenza and the influenza vaccine. The third part contained items assessing patients’ attitudes toward vaccinating against influenza. The final part assessed influenza vaccination habits and previous experiences (including having experienced side effects), complications of DM, intention to receive influenza vaccine, and barriers towards getting the influenza vaccine. 

The scores for knowledge and attitudes were computed based on patients’ answers to the designated questions. Each correct answer in the knowledge section was granted one point, and zero points were given for incorrect answers. Attitudes towards the influenza vaccine included a Likert-style response scale (strongly disagree, disagree, neutral, agree, or strongly agree), ranging from 5 points for “strongly agree” to 1 point for “strongly disagree”. We applied reverse scoring to questions with negative wording.

#### 2.2.2. Validation of Custom-Developed Questionnaire

Content validity was assessed through a panel of experts, which consisted of two endocrinologists, an academic professor with expertise in clinical pharmacy, and two clinical pharmacists. The questionnaire was initially adapted from a previous study conducted in Jordan among individuals with asthma [13], and it was initially in developed English to align with our English literature review. It was then customized to align with our specific sample. Following this, it was translated/back-translated into Arabic, which is the native language of Jordan. This translation process involved two independent translators, resulting in two comparable versions. A pilot study including 30 diabetics was conducted to ensure the questionnaire’s clarity and intelligibility for Jordanian participants. The data from the pilot study were excluded from the final analysis. To assess the internal consistency of the latent variables (DM knowledge, knowledge about influenza and its vaccine, and attitudes towards the influenza vaccine), Cronbach’s alphas were computed. A Cronbach’s alpha value above 0.5 was considered acceptable for the knowledge variables [14], as lower values are common in binary data [15]. For the attitude scale, an acceptable Cronbach’s alpha was determined to be 0.7 or above. The calculated Cronbach’s alpha values ranged from 0.53 to 0.79, all surpassing the 0.5 threshold.

#### 2.2.3. The Validated Arabic Version of Summary of Diabetes Self-Care Activities Measure (SDSCA)

The Summary of Diabetes Self-Care Activities Measure (SDSCA) [16] was applied in the present study to evaluate patients’ self-care practices in relation to managing their DM. The questionnaire consisted of 10 questions using an 8-point Likert scale, which measured the frequency of performing diabetes self-care activities over the past seven days, including diet, exercise, blood glucose testing, and foot care. The respondent marks the number of days for which the indicated behavior was performed; the questionnaire additionally includes a yes/no question on the respondent’s smoking habits. The total score of the SDSCA is computed based on the mean of the 11 items.

### 2.3. Sample Size Calculation

We utilized a convenience sampling method, considering a 50% population proportion, a 95% significance level, and a 5% margin of error to determine the minimum required sample size. This calculation indicated a minimum sample size of 385 participants. In our study, we included a total of 418 patients [17]. 

### 2.4. Statistical Analysis

SPSS version 28.0 was used to analyze the data. Continuous variables were represented as means and standard deviations, while categorical variables were represented as frequencies and percentages. Knowledge and attitude scores were computed by adding up the points assigned to specific items. A multinomial logistic regression model was constructed to assess variables associated with the intention to receive the influenza vaccine in the current year. The independent variables included in the model were age, DM duration, HbA1c level, DM knowledge, DM attitude, DM self-care practices, knowledge about flu and its vaccine, number of DM complications, gender, education, marital status, income level, and whether individuals had received the flu vaccine before. The significance level was determined at *p* < 0.05. 

## 3. Results

The total sample in the present study consisted of 418 diabetic adults (53.3% of whom were females) with a mean age of 49 (±14) years. The majority (84%) were married, and 78.2% were earning between 500 JOD and 1000 JOD monthly. The mean duration of DM diagnosis was 10 years (±8), with 70.1% of the patients diagnosed with type 2 DM. In addition, 58.6% of patients were non-smokers, 29.9% were smokers, and 11.5% were former smokers. The mean score for HbA1c was 7.79 (±1.58) (see Table 1). 

Table 2 represents diabetics’ responses to the DM knowledge items. The average DM knowledge score was 5.38 (±1.39) out of the highest achievable score of 7 (Cronbach’s Alpha = 0.53). The highest percentage of “yes” answers were observed for the items “Is diabetes a chronic disease?” and “Do you know how to properly use diabetes medications?” (96.2% and 95.5%, respectively). On the other hand, the item with the lowest percentage was “Do you know how to check if you have diabetic foot?” (45.2%).

Table 3 presents diabetics’ responses to the knowledge items regarding influenza and the influenza vaccine. The mean for the flu and flu vaccine knowledge score was 2.96 (±2.14) out of the highest achievable score of 8 (Cronbach’s Alpha = 0.79). The highest percentage of correct answers was observed for the item “Influenza can spread from one person to another”, followed by “Influenza is the same as the common cold” (85.9% and 62.2%, respectively). Conversely, the items with the lowest percentage were “When is the appropriate time to take the influenza vaccine?” (17.7%) and “Influenza is caused by bacteria” (17.9%).

Table 4 displays diabetics responses regarding the influenza vaccine attitude items. The average attitude score toward the influenza vaccine was 24.87 (±3.85) out of the highest achievable score of 40 (Cronbach’s Alpha = 0.74). On the reversed-coded statements, most diabetics (32.6%) “strongly disagreed/disagreed” with the statement “Catching influenza is not a problem for me”, while 18.2% “strongly disagreed/disagreed” with the statement “The influenza vaccination may cause complications/troubles for me”. For the remaining statements, 36.2% of participants “strongly agreed/agreed” with the statement “It is easy to reach the pharmacy/hospital to receive the influenza vaccination”, while 17.9% “strongly agreed/agreed” with the statement “My physician believes that I should receive the influenza vaccine”. 

The responses of diabetics to the practice items related to the influenza vaccine are provided in Table 5. The analysis showed that 70.6% of respondents had never received the influenza vaccine. The most-reported side effect among those who had received the vaccine before was headache. Nausea was the least frequently reported side effect (35.8% and 10.6%, respectively). In addition, 39.4% of the vaccinated diabetics reported having mild side effects after receiving the vaccine. When asked about their intention to take the influenza vaccine in the current year, 23.7% of respondents confirmed that they intended to do so, while 43.8% were unsure.

As shown in Table 6, 178 diabetics reported having one or more complications due to DM. The most-reported complications were eye problems and diabetic foot (25.8% and 19.1%, respectively), while the least-reported complications were kidney impairment (5.7%) and oral/gum issues (9.1%).

Diabetics’ responses to the DM self-care practice items are provided in Table 7. The mean for the total SDSCA score was 2.79 (±0.82) out of the highest achievable score of 11. The items with the highest mean were “On how many of the last SEVEN DAYS did you eat five or more servings of fruits and vegetables?”, “On how many of the last SEVEN DAYS did you eat high fat foods such as red meat or full-fat dairy products?”, “On how many of the last SEVEN DAYS did you test your blood sugar?”, and “On how many of the last SEVEN DAYS did you test your blood sugar the number of times recommended by your health care provider?” (4 ± 2 for all). On the other hand, the item with the lowest mean was “On how many of the last SEVEN DAYS did you inspect the inside of your shoes?” (1 ± 2).

A multinomial logistic regression was built to investigate the association between the intention to vaccinate against influenza and sociodemographic variables. The analysis revealed that as HbA1c increased, and rejection and hesitancy towards receiving the influenza vaccine decreased (OR = 0.666, 95% Cl (0.508–0.871), *p* = 0.003, and OR = 0.647, 95% Cl (0.519–0.808), *p* < 0.001, respectively). Moreover, as positive attitudes towards the influenza vaccine increased, rejection and hesitancy of receiving the influenza vaccine decreased (OR = 0.505, 95% Cl (0.424–0.601), *p* < 0.001 and OR = 0.729, 95% Cl (0.636–0.836), *p* < 0.001, respectively) (see Table 8).

As knowledge about flu and its vaccine increased, rejection of receiving the influenza vaccine decreased (OR = 0.627, 95% Cl (0.472–0.832), *p* = 0.001). Being married increased the odds of rejecting the influenza vaccine compared to non-married respondents (OR = 0.276, 95% Cl (0.077–0.986), *p* = 0.047). However, diabetic females had higher odds of rejecting the influenza vaccine compared to males (OR = 2.322, 95% Cl (1.052–5.125), *p* = 0.037) (see Table 8).

DM duration and income level were significantly associated with hesitancy toward receiving the influenza vaccine. As DM duration increased, hesitancy increased (OR = 1.053, 95% Cl (1.006–1.102), *p* = 0.028). Diabetics who were earning less than 500 JOD and who were earning between 500 JOD to 1000 JOD were more likely to exhibit hesitancy in receiving the influenza vaccine in contrast to those earning more than 1000 JOD (OR = 9.149, 95% Cl (1.014–82.572), *p* = 0.049 and OR = 5.442, 95% Cl (1.102–26.871), *p* = 0.038, respectively) (see Table 8).

Reasons for not wanting to receive the influenza vaccine among the patients who had no intention to receive the vaccine and those who were not sure are shown in Table 9. The reason most frequently cited for not getting vaccinated was “I don’t know the benefit of it” (42.6%), followed by “I forget it” (20.1%), whereas the least cited reasons were “It is expensive” (2.8%) and “The physicians do not recommend it” (3.8%).

## 4. Discussion

Compared to the general population, individuals with DM face a six-fold higher risk of hospitalization and a three-fold greater likelihood of mortality due to influenza-related complications [18]. Therefore, it is imperative to assess the knowledge, attitudes, and practices of diabetic patients regarding flu and its vaccination, which was the main objective of our study. The current study’s results demonstrated inadequate knowledge, unfavorable attitudes, and low willingness toward influenza vaccination among patients with diabetes. Factors such as HbA1c level, duration of diabetes, knowledge and attitudes toward the influenza vaccine, marital status, gender, and income showed significant and independent correlations with participants’ reluctance to receive the vaccine.

While the participants in the present study demonstrated good knowledge of diabetes, over half lacked knowledge on how to assess for the presence of diabetic foot (54.8%). Furthermore, participants exhibited limited knowledge concerning the flu and its vaccination, with the majority failing to identify the optimal timing for receiving the influenza vaccine (82.3%) and not knowing that influenza is not caused by bacteria (82.1%). This lack of knowledge about influenza and the vaccine is in line with previous Jordanian research involving patients with chronic conditions, some of whom had DM. [8] Comparatively, more favorable findings were reported in studies conducted in Saudi Arabia [19], China [20], and South Africa [21]. This underscores the importance of developing tailored educational strategies to enhance diabetic patients’ knowledge of diabetes, flu, and influenza vaccination in Jordan. These strategies should specifically target improving knowledge about diabetic foot care, understanding the optimal timing for the flu vaccine, and dispelling misconceptions about the causes of influenza.

In the current study, participants showed negative attitudes towards the influenza vaccine, with only 17.9% indicating that their physicians advised them to receive the vaccine, 18.2% expressing opposition regarding potential complications or problems associated with the vaccine, and 32.6% disagreeing with the idea that contracting influenza is not a concern for them. In a study conducted in Turkey, doctors’ recommendations emerged as the most influential factor in encouraging diabetic patients to receive the influenza vaccine [22]. This pattern of physician endorsement as a strong motivator for vaccination has also been observed in other studies [23,24]. A study conducted among patients with Type 1 and Type 2 DM patients in South Africa reported that nearly half held the belief that the influenza vaccine is safe and lacks significant side effects that inhibit its acceptance [21]. In Saudi Arabia, a study conducted on individuals visiting four primary healthcare centers in Riyadh reported that only 36.2% of the participants agreed that the influenza vaccine had serious side effects and should not be taken [19], while another study conducted with diabetic patients found that only 5.7% of participants agreed with this statement [25]. A lack of physician’s recommendations, in addition to mistaken beliefs regarding certain aspects of the influenza virus and its vaccine, may explain the negative attitudes toward the vaccine in the present study. This underscores the important role of physicians in encouraging vaccine uptake. It also emphasizes the need for educational campaigns to enhance awareness among diabetic patients in Jordan regarding influenza, its vaccination, and the risks of complications associated with not receiving the vaccine.

The influenza vaccination rates among diabetic patients in our study fell significantly short of expectations, as only 29.4% reported having received the influenza vaccine at any point, and merely 23.7% expressed a willingness to receive it in the current year. The most common reason associated with this low rate of vaccination was the lack of information about the benefits of the vaccine. Similar influenza vaccination rates were reported in previous research conducted on diabetic [21,26,27] and nondiabetic individuals [19,22]. According to a study conducted in Turkey among patients visiting an infectious diseases and clinical microbiology outpatient clinic, the rate of regular annual influenza vaccination was notably low, standing at a mere 10.3% among vaccinated individuals. The primary obstacle to influenza vaccination identified in this study was patients’ belief that they did not need the vaccine [22]. In a study conducted in the United Arab Emirates among healthcare professionals, around half of participants reported receiving the vaccine, but only 38.9% received it annually, potentially because they were uncertain about its effectiveness [28]. Another study conducted on Tunisian elderly individuals with chronic diseases reported that 19.4% of them were vaccinated during the 2018–2019 influenza season, but 64.7% expressed willingness to be vaccinated in the next season regardless of vaccination status in the 2018–2019 season. The most common reasons that prevented them from getting vaccinated were concerns about side effects and a belief that the vaccine was ineffective in averting influenza illness [29]. The results of our investigation and previous ones indicate a notable lack of awareness regarding the influenza vaccine, necessitating the need to introduce impactful educational interventions aimed at enhancing individuals’ understanding of the vaccine’s efficacy and benefits, addressing safety concerns, and consequently boosting influenza vaccination rates.

The current findings revealed that as HbA1c levels increased, the hesitancy and rejection of receiving the influenza vaccine decreased. This might occur as a result of the fact that people with poorly controlled diabetes are more likely to develop complications from infections such as influenza. As they become more aware of their own particular risk and the potential severity of the illness, their tendency to obtain the influenza vaccine will rise. On the other hand, participants with longer disease duration showed higher hesitancy toward receiving the vaccine than those who had lived with DM for a shorter period of time. People who have lived with DM for an extended period may have previously taken influenza vaccines and encountered adverse effects or viewed the vaccine as ineffective, which can lead to hesitancy. Additionally, elderly diabetics may have come across misinformation or misconceptions about vaccines, fostering skepticism regarding their safety and effectiveness. Furthermore, consistent with the results reported in earlier studies [19,21,30], participants in our study who showed a solid knowledge of and favorable attitudes toward the influenza vaccine were more likely to actually receive it. This association can be explained by the fact that these people are more inclined to value their health, comprehend the benefits of immunization, and be encouraged to take precautions to avoid contracting influenza, highlighting the importance of enhancing DM patients’ knowledge and attitudes toward the influenza vaccine through appropriate educational initiatives. In keeping with the findings of an earlier study carried out in Singapore [26], the current study similarly revealed that patients with higher incomes were significantly less reluctant to receive influenza vaccine compared to those with lower incomes. Additionally, compared to their peers, female participants and married participants were much more likely to reject the influenza vaccine, which emphasizes the importance of targeting these individuals in future health education programs aiming at increasing diabetic patients’ awareness of the influenza vaccine.

The present study has several limitations. The adoption of a cross-sectional design in this study prevents the ability to definitively establish cause-and-effect relationships. Moreover, since this study relied on a self-reported survey, there is a possibility of social desirability and recall biases, wherein individuals may offer responses they deem socially acceptable rather than an accurate reflection of their genuine beliefs and behaviors. However, self-report data are a commonly used and efficient method for studying knowledge, attitudes, and practices among diabetic adults in relation to health practices [31,32,33]. Furthermore, assurances of anonymity and confidentiality were provided to participants in order to encourage more honest and accurate responses. Still, future research could involve combining self-report data with objective sources of information to triangulate findings and improve the validity of the results.

## 5. Conclusions

The current study demonstrated poor knowledge, unfavorable attitudes, and poor practices regarding the influenza vaccine. Several factors were associated with participants’ hesitancy towards receiving the vaccine, including HbA1c level, DM duration, knowledge and attitudes toward the influenza vaccine, marital status, gender, and income. Future health education programs should emphasize the importance of the influenza vaccine in diabetic patients and eliminate its related concerns, particularly in female patients, married patients, patients with low income, and those with longer DM durations.

## Figures and Tables

**Table 1 vaccines-11-01689-t001:** Sociodemographic characteristics of the participants.

		Mean (±SD) or Frequency (%)
Age		49 (±14)
Gender	Female	223 (53.3%)
	Male	195 (46.7%)
Education	Elementary school	130 (31.1%)
	High school	120 (28.7%)
	Diploma	79 (18.9%)
	University degree	89 (21.3%)
Marital status	Married	351 (84%)
	Unmarried	67 (16%)
Monthly income	Less than 500 JOD	63 (15.1%)
	500–1000 JOD	327 (78.2%)
	More than 1000 JOD	28 (6.7%)
DM duration (years)		10 (±8)
DM type	Type 1	93 (22.2%)
	Type 2	293 (70.1%)
	Type 3 (pregnancy)	32 (7.7%)
HbA1c (%)		7.79 (±1.58)
Are you a smoker?	No	245 (58.6%)
	Former	48 (11.5%)
	Yes	125 (29.9%)
Are you exposed to passive smoking?	No	187 (44.7%)
	Yes	231 (55.3%)

**Table 2 vaccines-11-01689-t002:** Patients’ responses to DM knowledge items.

	Yes	No
	Frequency (%)	Frequency (%)
Do you know how to measure your blood sugar levels at home?	366 (87.6%)	52 (12.4%)
Do you know how to check if you have diabetic foot?	189 (45.2%)	229 (54.8%)
Are you aware that weight loss can reduce diabetes complications?	312 (74.6%)	106 (25.4%)
Is diabetes hereditary?	273 (65.3%)	145 (34.7%)
Is diabetes a chronic disease?	402 (96.2%)	16 (3.8%)
Do you know how to properly use diabetes medications?	399 (95.5%)	19 (4.5%)
Can you recognize the symptoms of low blood sugar?	308 (73.7%)	110 (26.3%)

Note. Correct answer is “yes”.

**Table 3 vaccines-11-01689-t003:** Patients’ reponses to the knowledge items regarding influenza and the influenza vaccine.

		Frequency (%)
Influenza is the same as the common cold.	No	158 (37.8%)
	Yes *	260 (62.2%)
Influenza can spread from one person to another.	No	59 (14.1%)
	Yes *	359 (85.9%)
Antibiotics can be used to treat flu.	Yes	316 (75.6%)
	No *	102 (24.4%)
Influenza is caused by bacteria.	Yes	343 (82.1%)
	No *	75 (17.9%)
Is there a vaccine against the flu?	No	245 (58.6%)
	Yes *	173 (41.4%)
Does the vaccine have side effects?	No	308 (73.7%)
	Yes *	110 (26.3%)
When is the appropriate time to take the influenza vaccine?	January–March	21 (5%)
	September–October *	74 (17.7%)
	November–December	37 (8.9%)
	I don’t know	286 (68.4%)
How many doses of the influenza vaccine are required to complete the full vaccination course?	1–2 *	85 (20.3%)
	>2	21 (5%)
	I don’t know	312 (74.6%)

* Indicate correct answer.

**Table 4 vaccines-11-01689-t004:** Diabetics responses regarding influenza vaccine attitude items.

	Strongly Disagree	Disagree	Neutral	Strongly Agree	Strongly Agree
	Frequency (%)	Frequency (%)	Frequency (%)	Frequency (%)	Frequency (%)
I believe that I must receive the influenza vaccination	10 (2.4%)	49 (11.8%)	237 (57%)	108 (26%)	12 (2.9%)
My physician believes that I should receive the influenza vaccine	6 (1.4%)	71 (17%)	266 (63.6%)	51 (12.2%)	24 (5.7%)
It is easy to reach the pharmacy/hospital to receive the influenza vaccination	6 (1.4%)	41 (9.8%)	219 (52.5%)	134 (32.1%)	17 (4.1%)
Influenza vaccination prevents infection by the influenza virus	2 (0.5%)	35 (8.4%)	281 (67.4%)	81 (19.4%)	18 (4.3%)
The influenza vaccination may cause complications/troubles for me *	7 (1.7%)	69 (16.5%)	217 (52%)	98 (23.5%)	26 (6.2%)
I believe that I get sick because of the influenza shot *	12 (2.9%)	104 (24.9%)	237 (56.8%)	49 (11.8%)	15 (3.6%)
Catching the influenza is not a problem for me *	52 (12.5%)	84 (20.1%)	188 (45.1%)	77 (18.5%)	16 (3.8%)
I am worried about the chances of contracting the influenza because of the influenza vaccine *	17 (4.1%)	92 (22.1%)	251 (60.2%)	44 (10.6%)	13 (3.1%)

* Indicates reverse-coded statements.

**Table 5 vaccines-11-01689-t005:** Patients’ responses to influenza vaccine practice items.

		Frequency (%)
Do you intend to take the influenza vaccine this year?	No	136 (32.5%)
	Not sure	183 (43.8%)
	Yes	99 (23.7%)
Did you ever receive the flu vaccine?	No	295 (70.6%)
	Yes	123 (29.4%)
Side effect severity	Mild	37 (39.4%)
	Moderate	30 (31.9%)
	Severe	27 (28.7%)
Reported side effects		
Fever		38 (30.9%)
Redness		40 (32.5%)
Fatigue		24 (19.5%)
Headache		40 (35.8%)
Nausea		13 (10.6%)
No symptoms		29 (23.6%)

**Table 6 vaccines-11-01689-t006:** DM complications.

	No	Yes
	Frequency (%)	Frequency (%)
Complications	240 (57.4%)	178 (42.6%)
Diabetic foot	338 (80.9%)	80 (19.1%)
Heart diseases	354 (84.7%)	64 (15.3%)
Neuropathy	376 (90%)	42 (10%)
Kidney impairment	394 (94.3%)	24 (5.7%)
Eye problems	310 (74.2%)	108 (25.8%)
Oral and gum issues	380 (90.9%)	38 (9.1%)

**Table 7 vaccines-11-01689-t007:** Patients’ responses to the SDSCA items.

	Mean (±SD)
How many of the last SEVEN DAYS have you followed a healthful eating plan?	3 (±2)
On average, over the past month, how many DAYS PER WEEK have you followed your eating plan?	3 (±2)
On how many of the last SEVEN DAYS did you eat five or more servings of fruits and vegetables?	4 (±2)
On how many of the last SEVEN DAYS did you eat high fat foods such as red meat or full-fat dairy products?	4 (±2)
On how many of the last SEVEN DAYS did you participate in at least 30 min of physical activity? (Total minutes of continuous activity, including walking).	3 (±2)
On how many of the last SEVEN DAYS did you participate in a specific exercise session (such as swimming, walking, biking) other than what you do around the house or as part of your work?	2 (±2)
On how many of the last SEVEN DAYS did you test your blood sugar?	4 (±2)
On how many of the last SEVEN DAYS did you test your blood sugar the number of times recommended by your health care provider?	4 (±3)
On how many of the last SEVEN DAYS did you check your feet?	2 (±2)
On how many of the last SEVEN DAYS did you inspect the inside of your shoes?	1 (±2)
Total SDSCA score	2.79 (±0.82)

**Table 8 vaccines-11-01689-t008:** Multinomial regression model between the intention to vaccinate against influenza in the current year and several sociodemographic characteristics.

		No vs. Yes				Not Sure vs. Yes			
		*p*-Value	OR	95% Confidence Interval for OR		*p*-Value	OR	95% Confidence Interval for OR	
				Lower Bound	Upper Bound			Lower Bound	Upper Bound
Age		0.930	0.998	0.958	1.040	0.105	0.972	0.938	1.006
DM duration		0.053	1.055	0.999	1.113	0.028	1.053	1.006	1.102
HbA1c		0.003	0.666	0.508	0.871	<0.001	0.647	0.519	0.808
DM knowledge score		0.902	0.983	0.746	1.294	0.820	1.028	0.811	1.303
Attitude score		<0.001	0.505	0.424	0.601	<0.001	0.729	0.636	0.836
Total SDSCA score		0.237	0.741	0.451	1.218	0.569	0.888	0.591	1.336
Knowledge about flu and its vaccine score		0.001	0.627	0.472	0.832	0.761	1.036	0.827	1.297
Number of DM complications		0.352	1.211	0.809	1.811	0.301	1.197	0.852	1.682
Gender	Female	0.037	2.322	1.052	5.125	0.323	1.413	0.712	2.801
	Male	.	.	.	.	.	.	.	.
Education	Elementary	0.339	0.530	0.144	1.949	0.491	1.508	0.469	4.854
	High school	0.522	0.666	0.192	2.313	0.984	1.012	0.321	3.187
	Diploma	0.089	0.318	0.085	1.189	0.252	0.500	0.153	1.636
	University degree	.	.	.	.	.	.	.	.
Marital status	Married	0.047	0.276	0.077	0.986	0.862	0.901	0.280	2.903
	Non-married	.	.	.	.	.	.	.	.
Monthly income	Less than 500 JOD	0.567	1.940	0.200	18.791	0.049	9.149	1.014	82.572
	500–1000 JOD	0.381	1.991	0.427	9.283	0.038	5.442	1.102	26.871
	More than 1000 JOD	.	.	.	.	.	.	.	.
Have received the influenza vaccine before	No	0.867	0.900	0.264	3.072	0.164	2.020	0.751	5.434
	Yes	.	.	.	.	.	.	.	.

**Table 9 vaccines-11-01689-t009:** Reasons for non-vaccination.

	Frequency (%)
It is expensive	9 (2.8%)
I don’t believe it is effective	58 (18.2%)
I forget it	64 (20.1%)
I think it may be harmful	44 (13.8%)
The physicians do not recommend it	12 (3.8%)
I don’t know the benefit of it	136 (42.6%)
I get the flu although I was previously vaccinated	15 (4.7%)
Not available	27 (8.5%)

## Data Availability

Available upon request.

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
