# Peer review of "Examining Influenza Vaccination Patterns and Barriers: Insights into Knowledge, Attitudes, and Practices among Diabetic Adults (A Cross-Sectional Survey)"

_vaccines, 2023, doi:10.3390/vaccines11111689_

Round 1

Reviewer 1 Report

Comments and Suggestions for Authors

This paper aims to explore diabetics' knowledge influenza and its vaccine, their adherence to practices that improve diabetes control and their attitudes and practices towards vaccinating against influenza, and evaluate the variables associated with their intention to receive the influenza vaccine in the current year in Jordan.

In the Introduction section, the authors pointed out that the prevalence of DM and pre-DM in Jordan is around 45%, as well as the current problems with the implementation of vaccination, especially against influenza.

Line 61: It is mandatory to add a new paragraph in which the procedures for vaccination of people with diabetes against influenza in Jordan should be described: what year did the vaccination start, is it mandatory, is the vaccination paid for, how much national vaccination coverage has been achieved, etc.

The Methods and Results sections described in detail the applied methodology and results.

The Discussion section presents the results of this paper in detail in relation to the results of other similar studies.

Line 311: It is mandatory to add a new paragraph in which the limitations of this study will be stated.

In the Conclusions section, the most important results are highlighted, which can represent an important basis for future research. 

Comments on the Quality of English Language

The quality of English language is appropriate. 

Author Response

Dear Editor,

The authors would like to thank the editorial team for their time and efforts and the reviewers for their time and thorough evaluation of the manuscript. We believe that we have addressed all the raised comments which substantially improved the quality of the manuscript.

Please see our detailed responses below

Reviewer(s)' Comments to Author:

Reviewer(1)

This paper aims to explore diabetics' knowledge influenza and its vaccine, their adherence to practices that improve diabetes control and their attitudes and practices towards vaccinating against influenza, and evaluate the variables associated with their intention to receive the influenza vaccine in the current year in Jordan.

In the Introduction section, the authors pointed out that the prevalence of DM and pre-DM in Jordan is around 45%, as well as the current problems with the implementation of vaccination, especially against influenza.

Line 61: It is mandatory to add a new paragraph in which the procedures for vaccination of people with diabetes against influenza in Jordan should be described: what year did the vaccination start, is it mandatory, is the vaccination paid for, how much national vaccination coverage has been achieved, etc.

Response: Kindly note that the following has been added to the introduction section “In Jordan, the influenza vaccine is available in community pharmacies to the public. It is not mandatory for any particular population group, including those with diabetes, and it is not provided free of charge. Additionally, it is not part of the national vaccination program. There are no official reports available on influenza vaccination coverage among Jordanians, and the country lacks awareness campaigns regarding the vaccine [11]. This highlights the need to comprehend the factors contributing to the low uptake of the influenza vaccine and the phenomenon of vaccine rejection/hesitancy. This understanding is essential for the planning, implementation, and evaluation of effective immunization pro-grams in Jordan, and may help healthcare providers develop health campaigns to increase vaccination against influenza among diabetics and thereby improve diabetes control.”

The Methods and Results sections described in detail the applied methodology and results.

Response: Thank you

The Discussion section presents the results of this paper in detail in relation to the results of other similar studies.

Response: Thank you

Line 311: It is mandatory to add a new paragraph in which the limitations of this study will be stated.

Response: Thank you for the suggestion. We have added a paragraph that addresses the limitations of the study. It reads as follows: “The present study has several limitations. The adoption of a cross-sectional design in this study prevents the ability to definitively establish cause-and-effect relationships. Moreover, since this study relied on a self-reported survey, there is a possibility of social desirability and recall biases, wherein individuals may offer responses they deem socially acceptable rather than an accurate reflection of their genuine beliefs and behaviors. However, self-report data is a commonly used and efficient method for studying knowledge, attitudes, and practices among diabetic adults in relation to health practices (e.g.  ). Furthermore, assurances of anonymity and confidentiality were provided to participants, in order to encourage more honest and accurate responses. Still, future research could involve combining self-report data with objective sources of information to triangulate findings and improve the validity of the results.

In the Conclusions section, the most important results are highlighted, which can represent an important basis for future research. 

Response: Thank you

Reviewer 2 Report

Comments and Suggestions for Authors

Dear authors, 

This study untitled « Examining Influenza Vaccination Patterns and Barriers: In-2 sights into Knowledge, Attitudes, and Practices among Diabetic 3 Adults” aims to examine the knowledge, attitudes, and practices of diabetic patients regarding the influenza vaccine in Jordania. This is an interesting study but some comments should be considered and particularly, the material and methods section should be more precise because actually the conception of the questionnaire and its validation remains unclear.

Introduction

1.     L56 « DM patients. [8] » Please put the point after the reference

2.     L58 “the COVID-19 pandemic[9]. » Please add one space before the “[“

3.     L69 “in the current year.. “ Please delete one point

4.     Please add epidemiologic data concerning the number of diabetics in Jordania…

Materials and methods

1.     Please report the data according to EQUATOR recommendations for survey studies that is to say according to the Consensus -Based Checklist for Reporting of Survey Studies (CROSS) 

2.     Did you obtain the ethics committee approval? 

3.     L83 “This questionnaire had been developed after extensive literature review and was translated from English into Arabic following its development.” Please clarify. Did you use a validated questionnaire in English that you validated in Arabic? If yes, explain the procedure of translation, … If not, explain why the questionnaire was prepared in English and translated.

4.     L101-109 “The Arabic translation of the questionnaire was validated using a forward-back translation by two independent translators.” Please clarify because the questionnaire can not be validated using the forward-back translation. The transcription, cross-cultural adaptation and validation must follow a process as the one recommended by the WHO for example. Please describe more precisely the different steps

5.     Please add the questionnaire in supplementary file

Author Response

Dear Editor,

The authors would like to thank the editorial team for their time and efforts and the reviewers for their time and thorough evaluation of the manuscript. We believe that we have addressed all the raised comments which substantially improved the quality of the manuscript.

Please see our detailed responses below

Reviewer(s)' Comments to Author:

Reviewer(2)

This study untitled « Examining Influenza Vaccination Patterns and Barriers: In-2 sights into Knowledge, Attitudes, and Practices among Diabetic 3 Adults” aims to examine the knowledge, attitudes, and practices of diabetic patients regarding the influenza vaccine in Jordania. This is an interesting study, but some comments should be considered and particularly, the material and methods section should be more precise because actually the conception of the questionnaire and its validation remains unclear.

Introduction

  1. L56 « DM patients. [8] » Please put the point after the reference

Response: We have made the requested change, and the point is now placed after the reference.

  1. L58 “the COVID-19 pandemic[9]. » Please add one space before the “[“

Response: We have made the requested change, and a space is now added before the “[“

  1. L69 “in the current year.. “ Please delete one point

Response: We have made the requested change, and one point is now deleted.

  1. Please add epidemiologic data concerning the number of diabetics in Jordania…

Response: The epidemiologic data regarding the prevalence of diabetes in Jordan were added as follows: "According to the National Center for Diabetes, Endocrinology, and Genetics, the prevalence of DM and pre-DM in Jordan is around 45%. In addition, DM prevalence in Jordan is higher than in other countries, globally and in the Middle East." Kindly note that this data is the most recent available as of 2023.

Materials and methods

  1. Please report the data according to EQUATOR recommendations for survey studies that is to say according to the Consensus -Based Checklist for Reporting of Survey Studies (CROSS)

Response: Thank you for the suggestion. We have reviewed the CROSS guidelines and made the necessary modifications to ensure that our data are reported in compliance with these recommendations in the manuscript. 

  1. Did you obtain the ethics committee approval? 

Response: Yes, ethics approval was obtained from the following institutions:

Al-Zaytoonah University of Jordan (Reference #22/23/2020–2021)

Jordan University of Science and Technology (Reference #2021/07)

Ministry of Health in Jordan (Reference #MOH/REC/2023/119)

Kindly note this information has been added to the manuscript as follows: “Ethical approval was granted by Al-Zaytoonah University of Jordan (Reference #22/23/2020–2021), Jordan University of Science and Technology (Reference #2021/07), and the Ministry of Health in Jordan (Reference #MOH/REC/2023/119).”

  1. L83 “This questionnaire had been developed after extensive literature review and was translated from English into Arabic following its development.” Please clarify. Did you use a validated questionnaire in English that you validated in Arabic? If yes, explain the procedure of translation, … If not, explain why the questionnaire was prepared in English and translated.
  2. L101-109 “The Arabic translation of the questionnaire was validated using a forward-back translation by two independent translators.” Please clarify because the questionnaire can not be validated using the forward-back translation. The transcription, cross-cultural adaptation and validation must follow a process as the one recommended by the WHO for example. Please describe more precisely the different steps

Response: To clarify, the questionnaire was initially adapted from a previously validated questionnaire used in a study conducted in Jordan among individuals with asthma. The decision to use an existing questionnaire in English as a starting point was made due to its relevance and alignment with our research objectives. We then customized it to suit the specific requirements of our study population in Jordan. Following this adaptation, we initiated the process of translating it into Arabic, the native language of our target population. A detailed explanation of the validation process has been added to the method section. It reads as follows: “Content validity was assessed through a panel of experts, which consisted of two endocrinologists, an academic professor with expertise in clinical pharmacy, and a clinical pharmacist. The questionnaire was initially adapted from a previous study conducted in Jordan among individuals with asthma, and it was initially in developed English to align with our English literature review. It was then customized to align with our specific sample. Following this, it was translated/back-translated into Arabic which is the native language of Jordan. This translation process involved two independent translators, resulting in two comparable versions. A pilot study including 30 diabetics was conducted to ensure the questionnaire’s clarity and intelligibility for Jordanian participants. The data from the pilot study were excluded from the final analysis. To assess the internal consistency of the latent variables (DM knowledge, knowledge about influenza and its vaccine, and attitudes towards the influenza vaccine), Cronbach’s alphas were computed.” Line (101-112)

  1. Please add the questionnaire in supplementary file

Response: Kindly note that the questionnaire has been included in the supplementary file.

Round 2

Reviewer 1 Report

Comments and Suggestions for Authors

Thanks for the opportunity to review the revised version of the paper `ID: vaccines-2678658`.

The authors satisfactorily responded to my comments regarding the Introduction section and the Limitations subsection of this study. The authors made appropriate corrections in the revised version of the paper, citing several new references.

I thank the authors for their responses to my comments. 

Comments on the Quality of English Language

The quality of English language is appropriate.  

Reviewer 2 Report

Comments and Suggestions for Authors

Thnak you for considering my comments